# MINIF2F-DAFNY: LLM-Guided Mathematical Theorem Proving via Auto-Active Verification

**Mantas Bakšys** [1 2]   **Stefan Zetzsche** [2]   **Olivier Bouissou** [3]   **Sean B. Holden** [1]

## Abstract

LLMs excel at reasoning, but validating their steps remains challenging. Formal verification offers a solution through mechanically checkable proofs. Interactive theorem provers (ITPs) dominate mathematical reasoning but require detailed low-level proof steps, while auto-active verifiers offer automation but focus on software verification. Recent work has begun bridging this divide by evaluating LLMs for software verification in ITPs, but the complementary direction—LLMs for mathematical theorem proving in auto-active verifiers—remains unexplored. We present MINIF2F-DAFNY, the first translation of the widely-used mathematical benchmark miniF2F to an auto-active verifier: Dafny. We find that Dafny's automation alone solves 39-44% of problems with empty proofs, whereas many require substantial proof guidance in ITPs. We evaluate 8 off-the-shelf LLMs on proof generation, with the best model (Claude Opus 4.6) achieving 62.7% cumulative pass@4 on the full test set, improving over the 38.9% empty-proof baseline by 23.8 percentage points. These results show that auto-active verification offers a complementary empirical setting for AI-assisted mathematical reasoning, where LLMs provide high-level guidance while SMT automation handles low-level details. Our benchmark and evaluation infrastructure are publicly available on GitHub.

*Table 1.* The AI-assisted formal verification benchmark landscape.

| | Pure Mathematics | Software Verification |
|---|---|---|
| **Interactive Theorem Provers** | **miniF2F**[1] **PutnamBench**[2] | **CLEVER**[3] **Verina**[4] |
| **Auto-Active Verifiers** | **MINIF2F-DAFNY** *(This Work)* | **DafnyBench**[5] **VerifyThis**[6] |

**References:** [1](Zheng et al., 2022), [2](Tsoukalas et al., 2024), [3](Thakur et al., 2025), [4](Ye et al., 2025), [5](Loughridge et al., 2024), [6](Deng et al., 2025).

## 1. Introduction

Automating reasoning is a grand challenge in artificial intelligence, requiring both creative insight and rigorous logical deduction. LLMs have demonstrated strong capabilities, yet verifying the correctness of their derivations remains difficult, particularly for complex problems where models are less reliable. Formal verification offers a solution: proofs written in specialized formal languages can be mechanically checked with complete certainty.

Two paradigms dominate formal verification. Interactive theorem provers (ITPs) such as Lean, Isabelle, and Rocq are based on type theory, where human experts or AI systems must construct explicit proof terms step-by-step, often using tactics to manipulate the proof state. Benchmarks like miniF2F (Zheng et al., 2022) and PutnamBench (Tsoukalas et al., 2024) evaluate AI capabilities in this domain. Specialized systems like Aristotle (Achim et al., 2025) and Seed-Prover (Chen et al., 2025b) have achieved gold medal performance at the 2025 International Mathematical Olympiad by generating formal proofs in Lean.

In contrast, auto-active verifiers like Dafny, Why3 (Filliâtre & Paskevich, 2013), F* (Swamy et al., 2016), and Verus (Lattuada et al., 2023) have no explicit proof object. Instead, users provide annotations—such as assertions, lemmas, and calculational proofs—that hint SMT solvers, which automatically handle reasoning about arithmetic, arrays, and other theories common in program verification. This enables push-button automation for program verification, requiring manual intervention only when automation fails. Benchmarks like DafnyBench (Loughridge et al., 2024) and VerifyThis (Deng et al., 2025) evaluate AI capabilities for software

Mantas Bakšys worked on this project during an internship at Amazon Web Services in London, UK. [1]University of Cambridge, Cambridge, UK [2]Amazon Web Services, London, UK [3]Amazon Web Services, Boston, USA. Correspondence to: Mantas Bakšys <mb2412@cam.ac.uk>, Stefan Zetzsche <stefanze@amazon.co.uk>.

*Proceedings of the 43rd International Conference on Machine Learning*, Seoul, South Korea. PMLR 306, 2026. Copyright 2026 by the author(s).

```
include "../definitions.dfy"
include "../library.dfy"

lemma imo_1959_p1(n: nat)
  requires 0 < n
  ensures gcd(21*n + 4, 14*n + 3) == 1
{}
```

*(a)* Dafny (no LLM)

```
import Mathlib

theorem imo_1959_p1
  (n : ℕ)
  (h₀ : 0 < n) :
  Nat.gcd (21*n + 4) (14*n + 3) = 1 := by
  have eq1 : (21 : ℕ) * n + (4 : ℕ) = ((7
    : ℕ) * n + (1 : ℕ)) + ((14 : ℕ) * n +
    (3 : ℕ)) * (1 : ℕ) := by ring
  conv =>
    lhs
    arg 1
    rw [eq1]
  rw [Nat.gcd_add_mul_left_left ((7 : ℕ) *
    n + (1 : ℕ)) ((14 : ℕ) * n + (3 : ℕ))
    (1 : ℕ)]
  rw [Nat.gcd_comm]
  have eq2 : (14 : ℕ) * n + (3 : ℕ) = (1 :
    ℕ) + ((7 : ℕ) * n + (1 : ℕ)) * (2 : ℕ
    ) := by ring
  conv =>
    lhs
    arg 1
    rw [eq2]
  rw [Nat.gcd_add_mul_left_left (1 : ℕ)
    ((7 : ℕ) * n + (1 : ℕ)) (2 : ℕ)]
  rw [Nat.gcd_one_left ((7 : ℕ) * n + (1 :
    ℕ))]
```

*(b)* Lean (SeedProver (Chen et al., 2025b))

*Figure 1.* Dafny and Lean proofs for IMO 1959 Problem 1.

verification in auto-active verifiers, showing that LLMs can synthesize effective proof hints from verifier error feedback.

These complementary strengths have led to a natural division: ITPs with their extensive mathematical libraries remain the primary choice for pure mathematics, while auto-active verifiers are primarily applied to software verification. The division creates challenges for safety-critical software that requires deep mathematical reasoning to prove correctness—such as cryptographic implementations or programs with probabilistic guarantees. For example, the SampCert framework for differential privacy (de Medeiros et al., 2025) was implemented in Lean to access existing measure theory libraries, despite requiring a custom monadic language to model imperative programs. Conversely, verifying cryptographic libraries like HACL* (Zinzindohoué et al., 2017) in auto-active tools required building number-theoretic founda-

tions from scratch, despite their natural support for program modeling and routine proofs.

Recent work has begun addressing this divide: interactive proof modes for Dafny (Ciobâcă et al., 2025), Dafny-style verifiers in Lean like Velvet (Gladshtein et al., 2026), and SMT-like automation tactics as `grind` in Lean attempt to bring the strengths of each paradigm to the other. However, from an AI perspective, benchmarks have primarily focused on one direction (see Table 1): CLEVER (Thakur et al., 2025) and Verina (Ye et al., 2025) evaluate AI capabilities for using ITPs in software verification. The complementary direction—evaluating AI capabilities for using auto-active verifiers in mathematical reasoning—remains unexplored.

To address this gap, we introduce MINIF2F-DAFNY, the first translation of the widely-used mathematical benchmark miniF2F (Zheng et al., 2022) to an auto-active verification language: Dafny. Previously, the benchmark existed only in interactive theorem provers (Lean, Isabelle, HOL Light, Metamath). This enables us to empirically study mathematical reasoning in an auto-active setting, where automated solvers handle low-level proof obligations and LLMs provide higher-level proof guidance.

We find that Dafny's automation verifies 95 of 244 test problems (38.9%) and 106 of 244 validation problems (43.4%) with empty proofs, requiring zero external input. For comparison, Lean's `grind` tactic, its most powerful SMT-style automation tactic, solves 79 of 244 miniF2F test problems (32.4%). Dafny therefore solves more test problems in this evaluation, but the systems also exhibit complementary automation profiles: of the 107 problems solved by at least one of Dafny or `grind`, 67 are solved by both, 28 only by Dafny, and 12 only by `grind`. The Dafny-only problems are concentrated in `mathd_algebra` and `mathd_numbertheory`. This suggests that Dafny's SMT-backed automation discharges a substantial fragment of miniF2F, while `grind`'s exclusive successes show that the two baselines capture different proof patterns.

We evaluate 8 off-the-shelf LLMs on providing proof hints to guide Dafny's automation. The best model we test, Claude Opus 4.6, achieves 62.7% cumulative pass@4 on the full test set, improving over the 38.9% verifier baseline by 23.8 percentage points. While this is still less than state-of-the-art results in ITPs like Lean, those approaches use specialized models or sophisticated agentic frameworks with substantial compute budgets, whereas we use general-purpose models with modest compute and unoptimized mathematical libraries. Moreover, our proofs are often significantly shorter and more human-readable (Figure 7).

These results demonstrate that an auto-active verifier exposes a distinct empirical setting for AI-assisted mathematical reasoning: SMT automation handles many low-level

```
method Max(a: int, b: int) returns (m: int)
  requires true
  ensures m >= a && m >= b
  ensures m == a || m == b
{
  if a >= b { m := a; } else { m := b; }
}
```

*Figure 2.* A simple Dafny method with specifications.

obligations directly, while LLMs focus on higher-level proof structure.

Our main contributions are as follows:

- We present MINIF2F-DAFNY, the first translation of miniF2F to an auto-active verification language, enabling empirical study of mathematical reasoning in a verifier designed around SMT-backed automation.
- We establish baseline results showing that Dafny's automation alone solves 39-44% of problems with no manual intervention.
- We evaluate 8 off-the-shelf LLMs on providing proof hints for problems requiring human guidance, achieving 62.7% cumulative pass@4 with our best model.
- We analyze a complementary proof-generation regime in which automated solvers handle low-level logical steps while LLMs focus on high-level proof structure.

**Conflict of Interest Disclosure.** This work was funded by Amazon Web Services. M.B. carried out this work during an internship at Amazon Web Services, and S.Z. and O.B. are employed by Amazon Web Services. S.B.H. is a Professor at the University of Cambridge.

## 2. The Dafny Language

We now describe Dafny's verification mechanism in detail. Understanding how its automation works and how users guide it through ghost code is essential for interpreting our experimental results.

### 2.1. Logical Foundations and Automation

Dafny is a verification-aware programming language based on Floyd-Hoare logic (Hoare, 1969). Programs are annotated with specifications in the form of Hoare triples $P s Q$, where $P$ is a precondition (`requires`), $Q$ is a postcondition (`ensures`), and $s$ is the program body. Figure 2 shows a simple example.

The verification mechanism relies on *Weakest Precondition* calculus. Dafny translates the high-level code and specifications into an intermediate representation (Boogie (Barnett et al., 2005)), which computes the weakest precondition $wp(s, Q)$—the necessary and sufficient condition for $s$ to terminate in a state satisfying $Q$. The system

then generates a Verification Condition (VC) of the form $P \implies wp(s, Q)$.

This VC is encoded into First-Order Logic and dispatched to an SMT solver, typically Z3 (De Moura & Bjørner, 2008). The solver attempts to determine the validity of the formula using decision procedures for specific theories, including linear arithmetic, arrays, and uninterpreted functions. If the solver determines the negation of the VC is unsatisfiable, the program is verified.

This architecture has enabled large-scale verification efforts across industry and academia. Applications include AWS's Cedar authorization engine (Hicks, 2023b), the AWS Cryptographic Material Providers Library (Hicks, 2023a), VMware's VeriBetrFS file system (Hance et al., 2020), Microsoft's IronFleet distributed systems (Hawblitzel et al., 2017), and ConsenSys's Ethereum Virtual Machine specification (ConsenSys, 2023). Notably, AWS rewrote and verified its policy authorization engine in Dafny, which is invoked 1 billion times per second (Chakarov et al., 2025). Despite this success in software verification, Dafny's application to pure mathematical reasoning remains unexplored.

### 2.2. Proof Construction: Ghost Code vs. Proof Terms

The fundamental difference between Dafny and ITPs lies in what constitutes a proof. In ITPs based on type theory (such as Lean or Rocq/Coq, which use variants of dependent type theory), a proof is an explicit syntactic term that inhabits a proposition's type, constructed step-by-step through tactics. In Dafny, there is no explicit proof object. Instead, proof construction is the process of guiding the SMT solver's search. When Z3 fails to verify a VC automatically, users must provide auxiliary information known as *proof hints* or *ghost code*. This code is erased during compilation and exists solely to guide verification.

Common forms of ghost code include:

- **Calculational Proofs:** The `calc` statement allows users to break a complex equality or inequality into a chain of transitive steps, each of which must be automatically verifiable by Z3.
- **Assertions:** Intermediate `assert P` statements serve as cut points during search, forcing the solver to prove a proposition $P$ before using it to prove the goal.
- **Lemmas:** Users can encapsulate difficult sub-goals into separate `lemmas` with their own specifications, manually instantiating them where needed.

This paradigm offers a key advantage: the SMT solver automatically handles low-level logical bookkeeping, arithmetic reasoning, and routine deductions that would require explicit tactics in ITPs. Users focus on high-level proof structure while automation manages the details. However, the SMT solver is opaque—verification failures provide limited diag-

```
function {:axiom} sum<T>(s: set<T>, f: T ->
    real): (p: real)
  ensures s == {} ==> p == 0.0
  ensures forall x | x in s :: p == f(x) +
    sum(s - {x}, f)
```

*Figure 3.* Example definition from `definitions.dfy`: sum replicating an inductive definition as postconditions.

nostic information compared to ITP proof states, and solver heuristics can be sensitive to syntactic variations.

Despite this opacity—or perhaps because of it—the division of labor between human guidance and automated reasoning presents an opportunity for AI assistance. Recent work has begun exploring this direction for software verification: DafnyBench (Loughridge et al., 2024) and Clover (Sun et al., 2024) evaluate LLMs on generating verified code and proof hints in Dafny (similar efforts exist for other auto-active languages, e.g., AutoVerus (Yang et al., 2025) for Verus). Some evidence suggests that models may be more effective at generating Dafny code than ITP proofs (Bursuc et al., 2025; Cao et al., 2025). However, these efforts have focused on software correctness rather than pure mathematical reasoning—the gap that MINIF2F-DAFNY addresses.

## 3. MINIF2F-DAFNY Benchmark

### 3.1. Overview

The MINIF2F-DAFNY benchmark consists of 488 mathematical problems (244 test, 244 validation) translated from the Lean miniF2F benchmark (Zheng et al., 2022). Each problem is formulated as a Dafny lemma with preconditions (`requires` clauses) and postconditions (`ensures` clauses), but an empty proof body. The task is to synthesize a proof that satisfies Dafny's verifier when the empty proof is insufficient. Problem statements translate naturally from Lean to Dafny, as seen in Figure 1.

Problems span multiple mathematical domains including algebra, number theory, inequalities, combinatorics, and analysis. Each split (test and validation) contains 45 AMC problems, 15 AIME problems, and 20 IMO problems, with the remaining problems drawn from undergraduate mathematics courses. The benchmark tests both the ability to provide high-level proof strategies (lemma invocations, case splits, induction) and low-level proof details (algebraic manipulations, inequality chains).

Two supporting files provide mathematical infrastructure. The `definitions.dfy` file (283 lines) axiomatizes 81 definitions of core mathematical structures not native to Dafny: integers, rationals, reals, and complex numbers with their operations and properties. The `library.dfy` file

```
function {:axiom} log(x: real): (l: real)
  ensures x > 1.0 <==> l > 0.0
  ensures 0.0 < x < 1.0 <==> l < 0.0
  ensures x == 1.0 <==> l == 0.0

function {:axiom} logb(b: real, x: real): (
    l: real)
  ensures x > 1.0 <==> l > 0.0
  ensures 0.0 < x < 1.0 <==> l < 0.0
  ensures x == 1.0 <==> l == 0.0
  ensures b != 1.0 && b > 0.0 && x > 0.0
    ==> logb(b, x) == log(x) / log(b)
```

*Figure 4.* Example definitions from `definitions.dfy`: `log` and `logb` specifying range behavior through postconditions.

(938 lines) contains 174 lemmas encoding standard mathematical facts covering exponentials, logarithms, trigonometric functions, complex numbers, and number theory. This layer is intended as a compact, explicit foundation for an empirical study of proof synthesis in Dafny, not as a foundational mathematical library comparable to Mathlib. To isolate the reasoning capabilities of the SMT solver from the retrieval capabilities of utilizing a massive library, we restricted the background theory to a compact set of axioms. In the public benchmark repository,[1] we have proved 128 of the 160 axiomatized lemmas in `library.dfy`, reducing the trusted lemma surface by 80% (see Section 3.4). This tests whether AI systems can develop proofs by leveraging first-principles thinking and Dafny's Hoare-logic style specifications and SMT solver automation.

### 3.2. Definitions

The benchmark's mathematical foundations are provided through `definitions.dfy`, which axiomatizes the minimal mathematical infrastructure needed to express all miniF2F problem statements in Dafny. Unlike Mathlib (mathlib Community, 2020), where mathematical objects are constructed from first principles, our axiomatization reflects Dafny's design philosophy of leveraging SMT solver capabilities for automated reasoning from a trusted library of definitions rather than requiring extensive proof libraries. The translation was human-led and human-reviewed, with the support layer cross-checked against standard mathematical formulations and Mathlib where applicable.

The definitions are organized across four numeric domains: integers, rationals, reals, and complex numbers. Each domain includes standard arithmetic operations and domain-specific functions. Integers support summation and product over finite sets, modular arithmetic, and divisibility predicates. Rationals are represented explicitly as numerator-denominator pairs with fraction arithmetic. Reals include transcendental functions (exponential, logarithm, trigono-

---

[1] https://github.com/dafny-lang/miniF2F

```
lemma {:axiom} logb_change_base(b1: real,
    b2: real, x: real)
  requires b1 > 0.0 && b1 != 1.0
  requires b2 > 0.0 && b2 != 1.0
  requires x > 0.0
  ensures logb(b1, x) == logb(b2, x) / logb
    (b2, b1)
```

*Figure 5.* Example lemma from `library.dfy`: the logarithm change of base formula.

metric), power operations, and mathematical constants like $\pi$. Complex numbers provide field operations, norm functions, and complex exponential.

The axiomatization employs Dafny's `:axiom` attribute to declare functions through their specifications rather than implementations. The specifications vary in complexity depending on the function's nature. For transcendental functions like `log` (Figure 4), we axiomatize basic range properties—for instance, specifying that the logarithm is positive when the argument exceeds 1, negative when between 0 and 1, and zero at 1. For recursive functions like `sum` (Figure 3), we replicate inductive definitions as postconditions, specifying behavior on the empty set and on non-empty sets by removing one element. Number-theoretic functions (`gcd`, `lcm`, `prime`), combinatorial functions (`choose`, `factorial`), and utility functions (`floor`, `ceil`, `abs`) are similarly axiomatized. These contracts provide sufficient semantic information for Dafny's SMT-backed automation to verify many problems with empty proofs. While the axiomatizations follow standard mathematical definitions and we are cautiously confident in their correctness, this pragmatic approach does come with the usual soundness considerations inherent to axiomatization.

### 3.3. Library

Complementing the definitions, `library.dfy` provides 174 lemmas encoding standard mathematical facts. The library was developed through a problem-driven approach: lemmas were identified by analyzing proof attempts to determine needed facts, then cross-checked against Mathlib (mathlib Community, 2020) theorems for soundness before integration. Of the 160 lemmas originally marked with `:axiom`, we proved 128 in Dafny: 58 were discharged by the verifier with empty proof bodies, and 70 were generated by an LLM and then checked by Dafny. The remaining axiomatized lemmas primarily serve as defining properties for otherwise opaque functions. The lemmas span several mathematical domains:

- **Exponential/Logarithm** (27): Multiplication, addition properties, change of base
- **Powers and Roots** (28): Integer and real exponentiation, power laws, square roots

- **Trigonometry** (41): Angle addition, periodicity, special angle values, Pythagorean identity, bounds on $\pi$
- **Complex Numbers** (20): Field axioms, norm properties, Euler's formula
- **Number Theory** (5): GCD commutativity, GCD-LCM product formula
- **Analysis** (11): Limit uniqueness, continuity properties, absolute value inequalities
- **Sequences and Finite Sums** (32): Sum and product properties over sequences and sets, conversions
- **Rationals** (10): Field properties of rational numbers

Each lemma is specified through preconditions and postconditions. For example, the change of base formula for logarithms (Figure 5) requires positive bases different from 1 and ensures the standard relationship between logarithms in different bases. Additionally, some mathematical properties—such as associativity and additivity—cannot be naturally encoded as postconditions on individual functions and must instead be stated as separate lemmas.

The library's scope (938 lines) is intentionally minimal compared to Mathlib's 2M lines, focusing on prerequisites for olympiad-level mathematics. This reflects a pragmatic balance: provide sufficient theory to express and solve miniF2F problems while testing how well AI systems can develop proofs from a compact foundation using SMT automation. The library is likely incomplete, and extending it with additional lemmas would likely improve evaluation results.

### 3.4. Soundness of the Axiomatization

To assess the sufficiency of the support library and the correctness of the translation, we performed three validation checks. First, we attempted formalization of 14 lemmas selected from three distinct domains: integer summation (`Int.Sum`), integer products (`Int.Prod`), and logarithms (`Log`). All 14 problems were successfully solved and verified using only the provided `definitions.dfy` and `library.dfy` files with natural proofs. Second, in the public benchmark, we reduced the trusted lemma surface in `library.dfy` by proving 128 of the 160 lemmas that originally used `:axiom`, leaving primarily the defining properties of opaque functions as axioms. Third, we performed an ablation of the empty-proof baseline: of the 201 test and validation problems solved by the Dafny verifier without proof hints, 142 (71%) still verify after removing the `definitions.dfy` include entirely. The remaining 59 use standard mathematical primitives such as `pow`, complex arithmetic, `abs`, `sqrt`, `gcd`/`lcm`, and `factorial`. These checks do not turn the support layer into a foundational formalization, but they make the trusted boundary explicit and provide evidence that the main baseline results are not an artifact of a large hidden library.

*Table 2.* Evaluation results on the MINIF2F-DAFNY test set. For comparison, we also present results of the Lean `grind` tactic.

| Model | #[1] | Pass@1 | @2 | @4 |
|---|---|---|---|---|
| Claude Opus 4.6 | 244 | 58.2 | 61.5 | 62.7 |
| Claude Sonnet 4.5 | 244 | 52.5 | 53.7 | 55.7 |
| Claude Sonnet 4 | 244 | 49.7 | 50.9 | 53.0 |
| Claude Haiku 4.5 | 244 | 42.2 | 43.9 | 43.9 |
| GPT-OSS 20B | 244 | 41.4 | 42.2 | 43.4 |
| Qwen 3 Coder 480B | 244 | 45.5 | 47.5 | 48.8 |
| Qwen 3 32B | 244 | 43.0 | 43.9 | 45.5 |
| Qwen 3 Coder 30B | 244 | 43.0 | 44.3 | 44.7 |
| Dafny Verifier | 244 | 38.9 | 38.9 | 38.9 |
| Lean `grind` | 244 | 32.4 | 32.4 | 32.4 |

## 3.5. Validation

A critical component of our benchmark is validating that generated solutions adhere to the original problem statement without weakening it. This prevents solutions from strengthening preconditions, weakening postconditions, or introducing unsound axioms—issues we have observed in other benchmarks such as DafnyBench (Loughridge et al., 2024), where weak evaluation scripts allowed such violations. Given a problem stated as a Dafny `lemma`, our validator rejects solutions that contain warnings or errors during verification, modify or remove original `requires` clauses, weaken or remove original `ensures` clauses, use the `:axiom` attribute to assume facts without proof, or use `assume` statements to bypass verification.

Our validation pipeline operates in two stages. First, we invoke the Dafny verifier and parse its JSON output to detect verification failures. Second, we parse each problem `lemma` into its signature, preconditions, and postconditions using an extraction pipeline. The validator compares original and generated specifications: `requires` clauses must match exactly (no additions or removals), while `ensures` clauses must be a superset of the original. Here, adding postconditions is permitted because the strengthened lemma still implies the original statement—one can derive the original lemma from the stronger version. We additionally scan for `:axiom` attributes and `assume` statements. Critically, we distinguish verification diagnostics by source file—we allow warnings from library files (`definitions.dfy`, `library.dfy`) containing `:axiom` attributes, but any warnings or errors from the problem file itself trigger rejection. When validation fails, we return structured feedback indicating which clauses were modified or which unsound constructs were used, enabling models to self-correct in subsequent iterations.

## 4. Evaluation

We evaluate MINIF2F-DAFNY on baseline Dafny automation and 8 off-the-shelf LLMs to assess the effectiveness of auto-active verification for automated mathematical reasoning. We first measure how many problems Dafny's SMT-backed automation can solve without external intervention, then evaluate whether modern LLMs can provide proof hints to guide verification on the remaining problems.

### 4.1. Empty Proof Baseline

Dafny's verifier, powered by Z3, verifies 95 of 244 test problems (38.9%) and 106 of 244 validation problems (43.4%) with empty proofs—requiring no human-engineering or LLM-calls. To ensure reproducibility, we run verification on each problem for 4 attempts with a 30-second timeout and use Dafny version 4.11.0. To contextualize this result, we evaluated Lean 4's `grind` tactic—its most comprehensive automation combining simplification, case splitting, and arithmetic reasoning—on the miniF2F test problems in Lean. The `grind` tactic solves 79 of 244 problems (32.4%), compared to Dafny's 95 of 244 (38.9%). An overlap analysis shows that these baselines are complementary: 67 problems are solved by both, 28 only by Dafny, and 12 only by `grind`. Thus Dafny solves more problems in this direct comparison, while `grind` still covers some problems Dafny misses. For example, IMO 1959 Problem 1 (Figure 1) requires a substantial proof in Lean but verifies with an empty proof in Dafny.

### 4.2. LLM-Guided Proofs

For problems where empty proofs fail, we evaluate 8 off-the-shelf LLMs on providing proof hints. All models are evaluated on the full 244-problem test set.

**Experimental Setup** Each problem receives up to $N = 4$ independent attempt trajectories with $E = 3$ error correction iterations after the initial attempt (temperature $T = 0.5$, 8192 token limit per response). Each conversation begins with a system prompt (B.1) and the problem statement. Once model output is received, in the case of failures, we extract error traces from Dafny's output and append them to the error correction prompt (B.2) and the conversation history for iterative refinement. We evaluate models via the AWS Bedrock API: Claude (Opus 4.6, Sonnet 4.5, Sonnet 4, Haiku 4.5), GPT-OSS 20B, and Qwen 3 (Coder 480B, 32B, Coder 30B). No models are fine-tuned.

**Results** Table 2 presents pass@4 results on the test set. Claude Opus 4.6 achieves the highest performance at

---

[1]In Table 2, # denotes problems attempted, including 95/244 problems solved by the Dafny verifier baseline.

```
include "../definitions.dfy"
include "../library.dfy"

lemma imo_1964_p2(a: real, b: real, c: real)
  requires 0.0 < a && 0.0 < b && 0.0 < c
  requires c < a + b && b < a + c && a < b + c
  ensures a*a*(b+c-a) + b*b*(c+a-b) + c*c*(a+b-c) <=
    3.0*a*b*c
{
  assert b + c - a > 0.0;
  assert c + a - b > 0.0;
  assert a + b - c > 0.0;

  var lhs := a*a*(b+c-a) + b*b*(c+a-b) + c*c*(a+b-c);

  calc {
    lhs;
  == a*a*b + a*a*c - a*a*a + b*b*c + b*b*a - b*b*b
    + c*c*a + c*c*b - c*c*c;
  == a*a*b + a*a*c + b*b*c + b*b*a + c*c*a + c*c*b
    - a*a*a - b*b*b - c*c*c;
  }

  var expr := a*a*a + b*b*b + c*c*c + 3.0*a*b*c
            - a*a*b - a*a*c - b*b*a - b*b*c - c*c*a - c*
    c*b;

  var sos1 := (a-b)*(a-b)*(a+b-c);
  var sos2 := (b-c)*(b-c)*(b+c-a);
  var sos3 := (c-a)*(c-a)*(c+a-b);

  assert sos1 >= 0.0;
  assert sos2 >= 0.0;
  assert sos3 >= 0.0;

  sos_identity(a, b, c);
  assert expr >= 0.0;
  assert 3.0*a*b*c - lhs >= 0.0;
}

lemma sos_identity(a: real, b: real, c: real)
  requires 0.0 < a && 0.0 < b && 0.0 < c
  ensures 2.0*(a*a*a + b*b*b + c*c*c + 3.0*a*b*c -
    a*a*b - a*a*c - b*b*a - b*b*c - c*c*a - c*c*b) ==
    (a-b)*(a-b)*(a+b-c) + (b-c)*(b-c)*(b+c-a) +
    (c-a)*(c-a)*(c+a-b)
{ /* proof omitted for brevity */ }
```

*Figure 6.* IMO 1964 inequality problem solved by Claude Sonnet 4 using an auxiliary sum-of-squares identity lemma.

62.7% cumulative pass@4, solving 153 of 244 problems and adding 58 problems beyond the empty-proof baseline. Claude Sonnet 4.5 reaches 55.7%, adding 41 problems beyond the baseline. Several other models cluster in the 43-50% range. Qwen and GPT-OSS models trained primarily on general code show limited Dafny-specific knowledge, often confusing Dafny syntax with Lean. In contrast, larger models in the Claude family demonstrate stronger familiarity with verification idioms in Dafny.

LLM-generated proofs exhibit a range of sophistication. At one end, Qwen3 Coder 30B solves an AMC 12 problem about prime products (Figure 7) with a concise parity argument: by asserting that 195 is odd while the relevant even numbers are even, enabling Dafny to automatically verify the result. At the other end, Claude Sonnet 4 solves a challenging IMO 1964 inequality problem (Figure 6) by introducing an auxiliary sum-of-squares identity lemma. The

```
include "../definitions.dfy"
include "../library.dfy"

lemma amc12_2000_p6(p: nat, q: nat)
  requires prime(p)
  requires prime(q)
  requires 4 <= p <= 18
  requires 4 <= q <= 18
  ensures p*q - (p+q) != 194
{
  assert 195 % 2 == 1;   // 195 is odd
  assert 4 % 2 == 0;     // 4 is even
  assert 6 % 2 == 0;     // 6 is even
  assert 10 % 2 == 0;    // 10 is even
  assert 12 % 2 == 0;    // 12 is even
  assert 16 % 2 == 0;    // 16 is even
}
```

*Figure 7.* AMC 12 prime product problem solved by Qwen3 Coder 30B using concise parity reasoning.

proof demonstrates sophisticated mathematical reasoning: it uses `calc` statements to algebraically manipulate the left-hand side, introduces three sum-of-squares terms that are non-negative by construction, and invokes the auxiliary lemma to establish the inequality. These examples illustrate that modern LLMs can generate both concise hints that leverage SMT automation and non-trivial proof strategies involving auxiliary lemmas and structured reasoning. Additional examples are provided in the appendix.

**Error Analysis** Analysis of unverified problems reveals three main categories of difficulty:

- **Verification brittleness**: Minor variations in assertion order or `calc` organization cause verification failure.
- **Limited Dafny training data**: Models struggle with language-specific idioms like `calc` statements and `ghost` variables, producing syntactically correct but semantically ineffective proofs.
- **Mathematical complexity**: Problems require mathematical facts not present in the library, necessitating theory development from scratch.

**Discussion** Our results demonstrate that Dafny's SMT-backed automation provides a strong baseline for olympiad mathematics, with modern LLMs capable of providing effective proof hints to extend this further. The top model reaches 62.7% cumulative pass@4 on the test set, a 23.8 percentage point absolute improvement over the 38.9% baseline. However, significant room for improvement remains, particularly in handling verification brittleness and expanding model familiarity with Dafny-specific proof patterns.

## 5. Related Work

### 5.1. Formal Mathematical Reasoning Benchmarks

Formal mathematical reasoning benchmarks provide automatic verification mechanisms through proof assistants, in contrast to informal benchmarks like MATH (Hendrycks et al., 2021) and GSM8K (Cobbe et al., 2021), which evaluate natural language mathematical reasoning without formal guarantees of correctness.

Other competition-style benchmarks include FIMO (Liu et al., 2023), which contains Lean formalizations of IMO shortlist problems and PutnamBench (Tsoukalas et al., 2024), which contains Lean problems from the William Lowell Putnam Mathematical Competition and represents a significantly higher level of difficulty. ProofNet (Azerbayev et al., 2023a) focuses on Lean exercises from undergraduate mathematics textbooks. LeanDojo (Yang et al., 2023) provides a dataset derived from Lean's mathlib library.

MiniF2F remains the most widely adopted benchmark due to its manageable size, diverse problem coverage, and multi-language implementations.

### 5.2. AI for Interactive Theorem Proving

Whole-proof approaches generate complete proofs that are iteratively refined until verified. GPT-f (Polu & Sutskever, 2020) pioneered this in Metamath, followed by Formal Statement Curriculum Learning (Polu et al., 2022) in Lean. More recent whole-proof systems include DeepSeek-Prover (Xin et al., 2024), Seed-Prover (Chen et al., 2025b), and Kimina-Prover (Wang et al., 2025).

Step-wise approaches construct proofs incrementally using tree search, including Hypertree Proof Search (Lample et al., 2022) (Lean and Metamath) and LLEMMA (Azerbayev et al., 2023b) (Lean). Leadership on benchmarks has alternated between whole-proof and step-wise paradigms.

Hybrid systems combine informal reasoning, decomposition, and formal verification. Draft, Sketch and Prove (Jiang et al., 2022) and LEGO-Prover (Wang et al., 2023) work in Isabelle, using natural language proofs converted to formal sketches. DeepSeek-Prover-V2 (Ren et al., 2025) similarly uses subgoal decomposition to structure formal proof search. These systems demonstrate that delegating low-level proof work is a broad and important theme; our contribution is to study the same pressure point in an auto-active verifier where SMT automation is built into the verification loop.

Agentic frameworks orchestrate multiple components for proof generation. Systems include COPRA (Thakur et al., 2023), Prover-Agent (Baba et al., 2025), ProofCompass (Wischermann et al., 2025), and HILBERT (Varambally et al., 2025) in Lean.

Domain-specific approaches like AlphaGeometry (Trinh et al., 2024) target specific problem types, such as IMO geometry problems.

Success rates have improved dramatically recently, with HILBERT achieving 99.2% on miniF2F in Lean and 70.0% on PutnamBench. Multiple systems achieved gold medal performance on IMO 2025, including Seed-Prover (Chen et al., 2025b) and Aristotle (Achim et al., 2025) with formal solutions, and systems from Google DeepMind and OpenAI with natural language solutions. The same progression has continued in formal undergraduate-level mathematical theorem proving, with recent models such as Seed-Prover 1.5 (Chen et al., 2025a) and Numina-Lean Agent (Liu et al., 2026) leveraging Lean to solve 11 and 12 problems out of 12, respectively, in the well-known Annual Putnam Competition held in 2025.

## 6. Future Work

**Dafny-Specific Training**   Current models exhibit limited exposure to Dafny syntax and verification idioms, often conflating Dafny with interactive theorem provers. A key limitation is their inability to judge Z3's capacity to automatically discharge subgoals. Pre-training on curated Dafny corpora and fine-tuning on verification tasks could improve familiarity with auto-active verification patterns and teach effective use of calc statements, assertion placement, and ghost variables.

**Agentic Architectures**   Analogous to Lean-based systems like Aristotle (Achim et al., 2025), Seed-Prover (Chen et al., 2025b), and HILBERT (Varambally et al., 2025), Dafny-specific agentic frameworks could orchestrate proof search, lemma synthesis, and iterative refinement, exploiting the synergy between program synthesis and formal verification.

**Learned Lemma Libraries**   LEGO-Prover (Wang et al., 2023) shows how models can extract, generalize, and cache successful proof strategies as reusable lemmas. Adapting this to Dafny could enable cumulative learning across problems, mirroring human mathematical practice.

## 7. Conclusion

MINIF2F-DAFNY represents the first exploration of auto-active verification for pure mathematical reasoning, a domain traditionally dominated by interactive theorem provers. Our results show that the underlying verification paradigm materially affects AI proof-generation benchmarks: SMT-backed automation can discharge a large routine fragment directly, and LLMs can extend this baseline by supplying proof structure. This does not make Dafny a replacement for Lean-style formal mathematics, whose expressiveness

and libraries are substantially richer. Rather, it highlights a complementary point in the design space. Looking forward, we anticipate convergence between auto-active verification and interactive theorem proving—recent developments like Lean's `grind` tactic, Loom's Velvet framework (Gladshtein et al., 2026) and Strata's unified platform of intermediate representations in Lean (Strata Org, 2025) already demonstrate this trend. Our work suggests a promising path toward accessible, AI-assisted formal verification that exploits this synergy.

## Impact Statement

This paper presents work whose goal is to advance the field of Machine Learning. There are many potential societal consequences of our work, none which we feel must be specifically highlighted here.

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

# A. Example Solutions

In this section, we present some representative proof solutions generated by LLMs to illustrate the range of proof strategies employed in MINIF2F-DAFNY. We choose examples demonstrating sophisticated mathematical reasoning with auxiliary lemmas to show that the combination of Dafny's verification combined with precise LLM output can produce complete solutions to IMO-level mathematical problems.

### A.1. Complete Proof of `imo_1964_p2`

The solution to the IMO 1964 problem 2 (shown truncated in Section 4) synthesized by Claude Sonnet 4 demonstrates a sophisticated proof strategy using sum-of-squares decomposition. The proof establishes a triangle inequality by showing that $3abc - \text{LHS}$ can be expressed as a sum of squared terms, each multiplied by positive factors derived from the triangle inequalities. This requires proving an auxiliary algebraic identity through systematic expansion and term collection. Using the auxiliary facts, a complete proof is verified by Dafny's SMT solver based backend.

```
include "../definitions.dfy"
include "../library.dfy"

lemma imo_1964_p2(a: real, b: real, c: real)
  requires 0.0 < a
  requires 0.0 < b
  requires 0.0 < c
  requires c < a + b
  requires b < a + c
  requires a < b + c
  ensures a*a*(b+c-a) + b*b*(c+a-b) + c*c*(a+b-c) <= 3.0*a*b*c
{
  // First, note that the triangle inequalities ensure all terms are positive
  assert b + c - a > 0.0;
  assert c + a - b > 0.0;
  assert a + b - c > 0.0;

  // We'll prove that 3abc - LHS >= 0 by showing it equals a sum of squares
  var lhs := a*a*(b+c-a) + b*b*(c+a-b) + c*c*(a+b-c);

  // Expand the LHS
  calc {
      lhs;
  == a*a*(b+c-a) + b*b*(c+a-b) + c*c*(a+b-c);
  == a*a*b + a*a*c - a*a*a + b*b*c + b*b*a - b*b*b + c*c*a + c*c*b - c*c*c;
  == a*a*b + a*a*c + b*b*c + b*b*a + c*c*a + c*c*b - a*a*a - b*b*b - c*c*c;
  }

  // Now compute 3abc - lhs
  calc {
      3.0*a*b*c - lhs;
  == 3.0*a*b*c - (a*a*b + a*a*c + b*b*c + b*b*a + c*c*a + c*c*b - a*a*a - b*b*b - c*c*c);
  == 3.0*a*b*c - a*a*b - a*a*c - b*b*c - b*b*a - c*c*a - c*c*b + a*a*a + b*b*b + c*c*c;
  == a*a*a + b*b*b + c*c*c + 3.0*a*b*c - a*a*b - a*a*c - b*b*a - b*b*c - c*c*a - c*c*b;
  }

  // We'll show this equals (1/2) times a sum of squares
  var expr := a*a*a + b*b*b + c*c*c + 3.0*a*b*c - a*a*b - a*a*c - b*b*a - b*b*c - c*c*a -
    c*c*b;

  // Define the sum of squares terms with explicit non-negativity proofs
  var diff_ab := a - b;
  var diff_bc := b - c;
  var diff_ca := c - a;

  // Squares are always non-negative
  assert diff_ab * diff_ab >= 0.0;
  assert diff_bc * diff_bc >= 0.0;
```

```
    assert diff_ca * diff_ca >= 0.0;

    // Triangle inequality factors are positive
    assert a + b - c > 0.0;
    assert b + c - a > 0.0;
    assert c + a - b > 0.0;

    var sos1 := diff_ab * diff_ab * (a + b - c);
    var sos2 := diff_bc * diff_bc * (b + c - a);
    var sos3 := diff_ca * diff_ca * (c + a - b);

    // Each SOS term is non-negative (product of non-negative terms)
    assert sos1 >= 0.0;
    assert sos2 >= 0.0;
    assert sos3 >= 0.0;

    // Prove the key identity: 2*expr = sos1 + sos2 + sos3
    sos_identity(a, b, c);

    // Therefore expr = (1/2)(sos1 + sos2 + sos3) >= 0
    assert expr >= 0.0;
    assert 3.0*a*b*c - lhs >= 0.0;
}

lemma sos_identity(a: real, b: real, c: real)
  requires 0.0 < a
  requires 0.0 < b
  requires 0.0 < c
  ensures 2.0*(a*a*a + b*b*b + c*c*c + 3.0*a*b*c - a*a*b - a*a*c - b*b*a - b*b*c - c*c*a -
    c*c*b)
      == (a-b)*(a-b)*(a+b-c) + (b-c)*(b-c)*(b+c-a) + (c-a)*(c-a)*(c+a-b)
{
  var lhs := 2.0*(a*a*a + b*b*b + c*c*c + 3.0*a*b*c - a*a*b - a*a*c - b*b*a - b*b*c - c*c*
    a - c*c*b);
  var rhs := (a-b)*(a-b)*(a+b-c) + (b-c)*(b-c)*(b+c-a) + (c-a)*(c-a)*(c+a-b);

  // Expand each term on the right-hand side systematically
  var term1_expanded := expand_first_term(a, b, c);

  var term2_expanded := expand_second_term(a, b, c);

  var term3_expanded := expand_third_term(a, b, c);

  // Sum all three expanded terms
  calc {
     rhs;
  == term1_expanded + term2_expanded + term3_expanded;
  == (a*a*a - a*a*b - a*b*b - a*b*c + 2.0*a*b*c + b*b*b - b*b*c) +
     (b*b*b - b*b*c - a*b*b - b*c*c + 2.0*a*b*c + c*c*c - a*c*c) +
     (c*c*c - a*c*c - b*c*c - a*a*c + 2.0*a*b*c + a*a*a - a*a*b);
  }

  // Collect like terms carefully
  calc {
     rhs;
  == 2.0*a*a*a + 2.0*b*b*b + 2.0*c*c*c + 6.0*a*b*c - 2.0*a*a*b - 2.0*a*b*b - 2.0*a*a*c -
    2.0*a*c*c - 2.0*b*b*c - 2.0*b*c*c;
  }

  // Use commutativity of multiplication
  assert a*b*b == b*b*a;
  assert a*c*c == c*c*a;
  assert b*c*c == c*c*b;

  calc {
```

```
      rhs;
   == 2.0*a*a*a + 2.0*b*b*b + 2.0*c*c*c + 6.0*a*b*c - 2.0*a*a*b - 2.0*b*b*a - 2.0*a*a*c -
      2.0*c*c*a - 2.0*b*b*c - 2.0*c*c*b;
   == 2.0*(a*a*a + b*b*b + c*c*c + 3.0*a*b*c - a*a*b - b*b*a - a*a*c - c*c*a - b*b*c - c*c*
      b);
   == lhs;
   }
}

function expand_first_term(a: real, b: real, c: real): real
{
  a*a*a - a*a*b - a*b*b - a*a*c + 2.0*a*b*c + b*b*b - b*b*c
}

lemma expand_first_term_correct(a: real, b: real, c: real)
  ensures (a-b)*(a-b)*(a+b-c) == expand_first_term(a, b, c)
{
  calc {
     (a-b)*(a-b)*(a+b-c);
  == (a*a - 2.0*a*b + b*b)*(a+b-c);
  == a*a*a + a*a*b - a*a*c - 2.0*a*a*b - 2.0*a*b*b + 2.0*a*b*c + b*b*a + b*b*b - b*b*c;
  == a*a*a - a*a*b - a*b*b - a*a*c + 2.0*a*b*c + b*b*b - b*b*c;
  == expand_first_term(a, b, c);
  }
}

function expand_second_term(a: real, b: real, c: real): real
{
  b*b*b - b*b*c - a*b*b - b*c*c + 2.0*a*b*c + c*c*c - a*c*c
}

lemma expand_second_term_correct(a: real, b: real, c: real)
  ensures (b-c)*(b-c)*(b+c-a) == expand_second_term(a, b, c)
{
  calc {
     (b-c)*(b-c)*(b+c-a);
  == (b*b - 2.0*b*c + c*c)*(b+c-a);
  == b*b*b + b*b*c - a*b*b - 2.0*b*b*c - 2.0*b*c*c + 2.0*a*b*c + b*c*c + c*c*c - a*c*c;
  == b*b*b - b*b*c - a*b*b - b*c*c + 2.0*a*b*c + c*c*c - a*c*c;
  == expand_second_term(a, b, c);
  }
}

function expand_third_term(a: real, b: real, c: real): real
{
  c*c*c - a*c*c - b*c*c - a*a*c + 2.0*a*b*c + a*a*a - a*a*b
}

lemma expand_third_term_correct(a: real, b: real, c: real)
  ensures (c-a)*(c-a)*(c+a-b) == expand_third_term(a, b, c)
{
  calc {
     (c-a)*(c-a)*(c+a-b);
  == (c*c - 2.0*c*a + a*a)*(c+a-b);
  == c*c*c + a*c*c - b*c*c - 2.0*a*c*c - 2.0*a*a*c + 2.0*a*b*c + a*a*c + a*a*a - a*a*b;
  == c*c*c - a*c*c - b*c*c - a*a*c + 2.0*a*b*c + a*a*a - a*a*b;
  == expand_third_term(a, b, c);
  }
}
```

*Listing 1.* Complete proof of `imo_1964_p2` using a sum-of-squares identity.

## A.2. Complete Proof of `algebra_amgm_sumasqdivbgeqsuma`

The `algebra_amgm_sumasqdivbgeqsuma` problem requires a proof of the algebraic inequality: $\frac{a^2}{b} + \frac{b^2}{c} + \frac{c^2}{d} + \frac{d^2}{a} \geq a + b + c + d$ for positive reals $a, b, c, d$. The solution demonstrates sophisticated proof engineering by reducing the problem to the Cauchy-Schwarz inequality. The proof, synthesized by Claude Sonnet 4, introduces auxiliary lemmas to establish the Cauchy-Schwarz inequality for four terms and uses a variable substitution ($x_i = \frac{a}{\sqrt{b}}$, $y_i = \sqrt{b}$, ...) to transform the target inequality into a form where Cauchy-Schwarz applies directly. This exemplifies how modern LLMs can identify complex proof strategies in Dafny and structure complex arguments through helper lemmas.

```
include "../definitions.dfy"
include "../library.dfy"

lemma cauchy_schwarz_four_terms(x1: real, x2: real, x3: real, x4: real, y1: real, y2: real
    , y3: real, y4: real)
  ensures (x1*y1 + x2*y2 + x3*y3 + x4*y4) * (x1*y1 + x2*y2 + x3*y3 + x4*y4) <=
          (x1*x1 + x2*x2 + x3*x3 + x4*x4) * (y1*y1 + y2*y2 + y3*y3 + y4*y4)
{

  var lhs := (x1*y1 + x2*y2 + x3*y3 + x4*y4) * (x1*y1 + x2*y2 + x3*y3 + x4*y4);
  var rhs := (x1*x1 + x2*x2 + x3*x3 + x4*x4) * (y1*y1 + y2*y2 + y3*y3 + y4*y4);
  var diff := rhs - lhs;
  var sum_of_squares := (x1*y2 - x2*y1)*(x1*y2 - x2*y1) +
  (x1*y3 - x3*y1)*(x1*y3 - x3*y1) +
  (x1*y4 - x4*y1)*(x1*y4 - x4*y1) +
  (x2*y3 - x3*y2)*(x2*y3 - x3*y2) +
  (x2*y4 - x4*y2)*(x2*y4 - x4*y2) +
  (x3*y4 - x4*y3)*(x3*y4 - x4*y3);

  assert sum_of_squares >= 0.0;
}

lemma division_by_sqrt_product(x: real, y: real)
  requires x > 0.0
  requires y > 0.0
  ensures x / (sqrt(y) * sqrt(y)) == x / y
{
  assert sqrt(y) * sqrt(y) == y;
}

lemma algebra_amgm_sumasqdivbgeqsuma(a: real, b: real, c: real, d: real)
  requires 0.0 < a
  requires 0.0 < b
  requires 0.0 < c
  requires 0.0 < d
  ensures (a*a)/b + (b*b)/c + (c*c)/d + (d*d)/a >= a + b + c + d
{

  var x1 := a / sqrt(b);
  var x2 := b / sqrt(c);
  var x3 := c / sqrt(d);
  var x4 := d / sqrt(a);

  var y1 := sqrt(b);
  var y2 := sqrt(c);
  var y3 := sqrt(d);
  var y4 := sqrt(a);

  assert sqrt(a) > 0.0;
  assert sqrt(b) > 0.0;
  assert sqrt(c) > 0.0;
  assert sqrt(d) > 0.0;

  division_by_sqrt_product(a*a, b);
```

```
  division_by_sqrt_product(b*b, c);
  division_by_sqrt_product(c*c, d);
  division_by_sqrt_product(d*d, a);

  cauchy_schwarz_four_terms(x1, x2, x3, x4, y1, y2, y3, y4);

  assert x1*x1 == (a/sqrt(b)) * (a/sqrt(b));
  assert x1*x1 == (a*a) / (sqrt(b) * sqrt(b));
  assert x1*x1 == (a*a) / b;

  assert x2*x2 == (b*b) / c;
  assert x3*x3 == (c*c) / d;
  assert x4*x4 == (d*d) / a;

  var sum_x_sq := x1*x1 + x2*x2 + x3*x3 + x4*x4;
  var target_lhs := (a*a)/b + (b*b)/c + (c*c)/d + (d*d)/a;
  assert sum_x_sq == target_lhs;

  assert x1*y1 == (a/sqrt(b)) * sqrt(b) == a;
  assert x2*y2 == (b/sqrt(c)) * sqrt(c) == b;
  assert x3*y3 == (c/sqrt(d)) * sqrt(d) == c;
  assert x4*y4 == (d/sqrt(a)) * sqrt(a) == d;

  var sum_xy := x1*y1 + x2*y2 + x3*y3 + x4*y4;
  assert sum_xy == a + b + c + d;

  assert y1*y1 == sqrt(b) * sqrt(b) == b;
  assert y2*y2 == sqrt(c) * sqrt(c) == c;
  assert y3*y3 == sqrt(d) * sqrt(d) == d;
  assert y4*y4 == sqrt(a) * sqrt(a) == a;

  var sum_y_sq := y1*y1 + y2*y2 + y3*y3 + y4*y4;
  assert sum_y_sq == a + b + c + d;

  assert sum_xy * sum_xy <= sum_x_sq * sum_y_sq;
  assert (a + b + c + d) * (a + b + c + d) <= target_lhs * (a + b + c + d);

  assert a + b + c + d > 0.0;
  assert (a + b + c + d) <= target_lhs;
}
```

*Listing 2.*

Complete proof of `algebra_amgm_sumasqdivbgeqsuma` using the Cauchy-Schwarz inequality.

# B. Prompts

## B.1. Initial Proof Synthesis Prompt

We craft an initial model prompt to establish the task context and constraints that we present below. It emphasizes preserving problem specifications, prohibits unsound constructs (axioms, assumptions), and encourages strategic use of Dafny idioms like calc statements and assertions. The prompt explicitly instructs models to prove any mathematical results they invoke, preventing reliance on unstated *"classical theorems."* This design reflects our goal of testing proof synthesis rather than library lookup.

```
# ROLE #
You are an expert Dafny programmer specializing in formal
verification, proof construction, mathematical reasoning and
proof hint infilling.

# OBJECTIVE #
Complete the proof body of the given incomplete Dafny lemma
to make it verify successfully. The lemma currently has an
empty body `{}` that you must replace with a complete,
```

```
verifying proof.

# AVAILABLE RESOURCES #

## definitions.dfy file content:
{definitions_file_content}

## library.dfy file content:
{library_file_content}

# TASK REQUIREMENTS #

You will receive an incomplete Dafny file containing a lemma
with:
- A complete signature (name and parameters)
- Complete `requires` clauses (preconditions)
- Complete `ensures` clauses (postconditions)
- An empty body `{}` that needs to be filled

Your task is to replace the empty body with a complete proof
that satisfies all postconditions.

# CRITICAL RULES #

1. PRESERVE SPECIFICATION: You MUST preserve the exact lemma
   signature and all `requires` clauses. Do NOT add, remove,
   or modify any `requires` clauses.

2. ENSURES CLAUSES: The `ensures` clauses MUST all be
   preserved. You MAY add additional `ensures` clauses if
   they may aid with verification but this is not encouraged.

3. COMPLETE IMPLEMENTATION: Provide a complete proof body.
   This may include:
   - Assertions to guide the verifier
   - Calc statements for step-by-step reasoning
   - Calls to helper lemmas (from library.dfy or your own)
   - Loop invariants
   - Case analysis or proof by contradiction

4. USE AVAILABLE RESOURCES: Leverage functions and lemmas
   from definitions.dfy and library.dfy without redefining
   them. They are already imported via
   `include "../definitions.dfy"` and
   `include "../library.dfy"`.

5. DEFINE HELPERS IF NEEDED: If you need helper lemmas or
   functions not available in the provided files, define
   them within your solution.

6. NO ASSUME OR AXIOM ATTRIBUTES: You may NOT use the
   `assume` statement or `{:axiom}` attribute anywhere in
   your solution. All auxiliary lemmas MUST include complete
   proofs.

7. PROVE ALL CLAIMS: If you reference mathematical results,
   theorems, or inequalities (e.g. Cauchy-Schwarz inequality,
   AM-GM, Schur's Inequality etc.) that are not present in
   the definitions.dfy or library.dfy files, you MUST prove
   them yourself within your solution. Do NOT rely on
   "classical results" or "known theorems" without providing
   a complete formal proof in Dafny.

8. EXPLICIT TRIGGERS FOR QUANTIFIERS: You MUST add explicit
   triggers to all quantifiers (forall/exists) where they
   are not automatically synthesized by Dafny. Use the
   `{:trigger}` attribute to specify triggers that help
   guide the verifier's instantiation of quantified variables.

# PROOF STRATEGIES #

Consider these approaches when constructing your proof:
- Direct proof: Use assertions and calc statements to show
  the postcondition follows from preconditions
- Calc statements: Use `calc` blocks for step-by-step
  algebraic calculations and equational reasoning - this is
  especially useful for mathematical proofs involving
  arithmetic, logarithms, or algebraic manipulations
- Case analysis: Split the proof into cases based on
  conditions
- Induction: For properties involving natural numbers or
```

```
    recursive structures
- Contradiction: Assume the negation and derive a
  contradiction
- Helper lemmas: Break complex proofs into smaller,
  manageable pieces
- Verifier hints: Add strategic assertions to guide Dafny's
  automated reasoning

The complete file must include:
- The original `include` statement
- The complete lemma with your proof body
- Any helper lemmas or functions you define

# INPUT FORMAT
```dafny
<INCOMPLETE_FILE_CONTENT>
```

# RESPONSE FORMAT #

Reason step-by-step first and then, provide the complete
Dafny file in this exact format:
```dafny
<COMPLETE_FILE_CONTENT>
```

---

```dafny
{incomplete_dafny_file_content}
```
```

**Listing 3.** System prompt for initial proof generation.

## B.2. Error Correction Proof Synthesis Prompt

When verification fails, the error correction prompt presented below provides structured feedback including Dafny's error messages and a reminder of the original specification. This enables iterative refinement while maintaining specification adherence. We keep the error-correction prompt brief for the purposes of reducing tokens used and maximizing information efficiency by avoiding repetition.

```
Your code failed Dafny verification with the following errors:

{errors}

Reminder - the original lemma you need to complete is:
```dafny
{incomplete_dafny_file_content}
```

Please fix these errors and provide the complete corrected
Dafny file. CRITICAL:
- You MUST preserve the exact lemma signature and all
  `requires` clauses from the original lemma above
- You MUST keep all `ensures` clauses from the original
  lemma above
- Only the proof body should be modified - do NOT change
  the specification
- Do NOT use `assume` statements or `{:axiom}` attributes
- Add explicit triggers to quantifiers where needed

Provide the corrected code in this format:
```dafny
<COMPLETE_CORRECTED_FILE_CONTENT>
```
```

**Listing 4.** Error correction prompt.

