# OpenReview forum: "miniF2F-Dafny: LLM-Guided Mathematical Theorem Proving via Auto-Active Verification"
_ICML.cc/2026/Conference — ICML 2026 regular_

### Official Review · Reviewer_RfdE · 2026-03-09

**Soundness:** 3
**Presentation:** 3
**Significance:** 2
**Originality:** 3
**Overall Recommendation:** 4
**Confidence:** 3

**Summary:**

This paper presents MINIF2F-DAFNY, a Dafny translation of miniF2F for studying LLM-guided mathematical theorem proving in an auto-active verification setting. The paper adapts miniF2F problems into Dafny, introduces a small supporting mathematical library, and defines a validation protocol intended to prevent trivial unsound shortcuts such as weakening specifications or adding assumptions. Empirically, the paper shows that Dafny alone already solves a substantial fraction of the translated benchmark, and that LLMs can further improve results by generating proof hints, assertions, calculations, and auxiliary lemmas. The main takeaway is a division of labor: SMT automation handles much of the low-level reasoning, while LLMs provide higher-level proof structure.

**Compliance With Llm Reviewing Policy:**

Affirmed.

**Final Justification:**

The rebuttal satisfactorily addressed both main concerns by clarifying the empirical scope of the axiomatization and confirming that the evaluation in Table 2 is complete, and the paper's originality and artifact contribution are sufficient to merit acceptance.

**Key Questions For Authors:**

See the questions in the Weaknesses section above.

**Limitations:**

yes

**Strengths And Weaknesses:**

**Strengths**

1. The paper studies a genuinely interesting and underexplored setting. Most prior work focuses on either ITPs for mathematics or auto-active verification for software; this paper targets the relatively missing field of auto-active verification for mathematical theorem proving.
2. Translating miniF2F into Dafny is a meaningful contribution. It requires adapting problem representations, building a mathematical support layer, and defining an evaluation protocol suited to Dafny.
3. The main empirical observation is interesting. The fact that Dafny solves many translated problems with empty proofs suggests that a nontrivial part of the proof burden in miniF2F-style tasks can be discharged automatically once low-level reasoning is delegated to SMT. In this setting, LLMs do not need to synthesize full low-level proofs, but can instead serve as generators of useful proof guidance.

**Weaknesses**

1. The main weakness is the heavy reliance on axiomatization. For this benchmark to be convincing, the background theory should be reduced to a minimal, auditable axiomatic core, and the remaining mathematical facts should be obtained through explicit proofs. In the current version, too many conclusions are packaged axiomatically, so it is hard to assess whether the reported results reflect real theorem proving progress or simply the strength of the assumed library.
3. The experimental section is incomplete. The paper explicitly states that some evaluations are still work in progress. If the authors complete these results, I would consider revising my score accordingly.

---

> ### Author Rebuttal · Authors · 2026-03-30
>
> We thank the reviewer for the constructive assessment and for recognizing the important and relevant setting our work explores.
>
> **On axiomatization.** We want to clarify what we are and are not claiming. Our paper is best read as an empirical study, not as the introduction of a foundational mathematical library for Dafny. The axiomatization exists to make miniF2F expressible in Dafny so we can study how much of it falls within reach of SMT automation and how effectively LLMs can extend that. We are not arguing that Dafny should replace Lean for mathematics.
>
> To ensure the experiment is meaningful, the benchmark was manually translated from Lean by human experts. Each axiomatized definition and lemma corresponds to a standard mathematical statement, cross-checked against Mathlib where applicable. For example, our `logb_change_base` lemma (Figure 5) was manually translated from `Real.div_logb` in Mathlib. The axiomatization is compact ($\sim$255 items vs.\ Mathlib's 2M+ lines), explicit, and publicly auditable. We manually verified 14 axiomatized lemmas by constructing full Dafny proofs (Section 3.4), and the validator (Section 3.5) prevents generated solutions from introducing `{:axiom}` attributes, `assume` statements, or weakening specifications. We are happy to reduce the axiomatization further by proving additional library lemmas constructively in the revision.
>
> **On incomplete evaluation.** The sentence about evaluation being work in progress is leftover from an earlier draft and should not have appeared in the submitted version. We apologize for the confusion. The evaluation in Table 2 is complete: all 7 models were evaluated on the full 244-problem test set. We will remove the outdated language in the revision.

---

> > ### Author Rebuttal · Reviewer_RfdE · 2026-04-04
> >
> > Thanks for your detailed rebuttals and I will raise my score to 4.

---

> > > ### Author Response · Authors · 2026-04-08
> > >
> > > We thank the reviewer for the updated assessment.
> > >
> > > As a brief follow-up: we have since proved 128 of the 160 axiomatized lemmas in `library.dfy`, reducing the trusted axiom surface by 80%, conducted an ablation confirming that the baseline results are robust to the design of `definitions.dfy`, and evaluated Claude Opus 4.6 on the test set (cumulative 62.7% pass@4, up from 55.7% with Sonnet 4.5). We will include all of these in the revision.

---

### Official Review · Reviewer_nWpJ · 2026-03-11

**Soundness:** 2
**Presentation:** 3
**Significance:** 1
**Originality:** 2
**Overall Recommendation:** 3
**Confidence:** 4

**Summary:**

This paper introduces miniF2F-Dafny, a Dafny version of the miniF2F formal mathematics benchmark consisting of 244 problems. The authors show that Dafny’s built-in automation alone can solve 39–44% of the problems without additional proof guidance. When large language models are used to generate proof hints that guide the verifier, the success rate increases to 55.7% with the strongest model.

**Compliance With Llm Reviewing Policy:**

Affirmed.

**Final Justification:**

The motivation presented in the paper is not entirely convincing to me. Using Dafny as the primary tool for expressing and verifying mathematics seems less comprehensive than existing systems like Lean for a wide range of reasons, which makes the overall scope feel quite niche. While the experimental results are reasonable, they do not seem to offer substantially new insights. In the rebuttal, the authors mention that they plan to revise the paper and include more comparisons, which aligns with my initial assessment.

**Key Questions For Authors:**

What is the main advantage of using Dafny instead of Lean for formalizing mathematical reasoning, especially considering that many non-trivial mathematical statements go beyond first-order logic?

**Limitations:**

yes

**Strengths And Weaknesses:**

### Strength
- The paper is well-written and easy to follow.
- The proposed Dafny version of miniF2F requires only a relatively small library and benefits from strong SMT-based automation, making it easier to solve some problems compared to approaches relying on Lean automation such as the `grind` tactic.
- The experiments show that LLM-provided hints can further assist the verifier and improve the overall automation performance.

### Weakness
- The motivation of the paper does not convince me. Dafny is primarily limited to first-order logic (+ theories), which is, in principle, less expressive for mathematics. In practice, the Dafny ecosystem is mainly designed for program verification, where this design choice makes more sense. In contrast, Lean has a much richer mathematical library and supports significantly stronger expressiveness through dependent type theory. Lean also provides several FOL-style automation tools such as Lean-SMT, Lean-Auto, and LeanHammer, which connect Lean statements to external ATP solvers. Moreover, Dafny proofs are not structured proofs like in Lean. From this perspective, it is unclear what the specific advantage of using Dafny is for mathematical reasoning, and the choice of Dafny as the main framework feels somewhat unconvincing.
- The provided examples suggest that the approach mainly handles relatively simple formula-based problems. It is unclear whether the framework can scale or generalize to more complex mathematical concepts or problems requiring deeper mathematical structures.
- The evaluation is relatively limited. It does not compare against Lean-SMT, Lean-Auto, or LeanHammer.

---

> ### Author Rebuttal · Authors · 2026-03-30
>
> We thank the reviewer for the candid review.
>
> **On the motivation for using Dafny.** We agree that the paper should not argue Dafny is a better framework for formal mathematics than Lean. It is not, for all the reasons the reviewer identifies: Lean has richer expressiveness, a vastly larger library, and stronger community support.
>
> Our paper is best read as an empirical study showing that performance on AI reasoning benchmarks depends substantially on the underlying verification tooling, not just the models. The concrete finding is that $\sim$39\% of miniF2F requires zero proof guidance in Dafny, while the same problems often require substantial proof construction in ITPs. Our overlap analysis supports this: of the 107 problems solved by at least one system, 28 are solved only by Dafny and 12 only by `grind`, with the Dafny-only problems concentrated in `mathd_algebra` and `mathd_numbertheory`. This suggests that a substantial fraction of the proof burden in miniF2F-style benchmarks is routine reasoning that ITPs currently require models to handle explicitly.
>
> This is relevant to the Lean community as well: the ongoing investment in SMT-style automation (`grind`, Velvet/Loom) reflects exactly the recognition that low-level automation matters. Our work provides empirical evidence for why that investment is worthwhile. The benchmark was manually translated from Lean by human experts, with each axiomatized item corresponding to a standard mathematical statement cross-checked against Mathlib ($\sim$255 items vs.\ Mathlib's 2M+ lines). We are not claiming to replicate Mathlib's depth, just enough to run the experiment.
>
> We will revise the paper to present this as the main motivation, rather than as a claim that Dafny is broadly preferable for mathematical formalization.
>
> **On scalability to complex mathematics.** We acknowledge that the current benchmark focuses on olympiad-style problems largely expressible in first-order logic with arithmetic theories. Problems requiring deeper mathematical structures (abstract algebra, topology, measure theory) would require richer type systems and libraries beyond what Dafny currently offers. We will state this limitation explicitly.
>
> **On missing comparisons.** We compare against Lean's `grind` tactic as it is the most comprehensive single built-in automation tactic. We acknowledge that Lean-SMT, Lean-Auto, and LeanHammer provide additional automation capabilities and that a comparison against these tools would strengthen the paper. We expect that combining multiple Lean automation tools would narrow the gap with Dafny's baseline, which would itself be an interesting finding about the convergence of the two paradigms. We will note this as a limitation and, if feasible, include additional comparisons in the revision.

---

> > ### Author Rebuttal · Reviewer_nWpJ · 2026-04-02
> >
> > I still think that using Dafny as the main tool for expressing and verifying mathematics is less compelling compared to Lean, which makes the scope of the paper feel somewhat niche. That said, I understand the authors’ argument that the work can be viewed as an empirical study of verifying mathematics across different verification tools, which does make some sense, and I have adjusted my rating accordingly. I would encourage the authors to revise the paper to clarify and better emphasize this positioning.

---

> > > ### Author Response · Authors · 2026-04-08
> > >
> > > We thank the reviewer for the updated assessment and for acknowledging the empirical study framing.
> > >
> > > As an additional update: we have conducted an ablation confirming that the baseline results are robust to the design of `definitions.dfy`, reduced the axiom surface in `library.dfy` by 80%, and evaluated Claude Opus 4.6 on the test set (cumulative 62.7% pass@4, up from 55.7% with Sonnet 4.5). We will integrate these results into the revision along with the clarifications from the rebuttal.

---

### Official Review · Reviewer_CXje · 2026-03-11

**Soundness:** 2
**Presentation:** 3
**Significance:** 2
**Originality:** 3
**Overall Recommendation:** 4
**Confidence:** 2

**Summary:**

This paper creates a benchmark miniF2F-Dafny, which is based on the questions from miniF2F, but translate the lean statements into Dafny. The authors find that Dafny’s automation alone solves about 40% of problems even with empty proofs (Dafny automatically verified them). They also test the ability of current LLMs on this benchmark.

**Compliance With Llm Reviewing Policy:**

Affirmed.

**Ethical Review Concerns:**

How do you justify that definitions.dfy is an appropriate foundational layer for this benchmark? More concretely, how sensitive are the reported results to the particular design of definitions.dfy? For example, if this file were modified in nontrivial ways, would the benchmark results change substantially?

**Final Justification:**

The authors partially addressed my concerns, so I will raise my score to 4.

**Key Questions For Authors:**

How can you justify that "definitions.dfy" is appropriate? To be specific, if you modify the "definitions.dfy", will the result dramatically change?

**Limitations:**

Yes

**Strengths And Weaknesses:**

Strength

Presentation: The paper is generally well presented. It clearly motivates the need for a non-ITP benchmark for mathematical reasoning, which is relatively underexplored. The paper also does a good job explaining the role of Dafny in this setting and how the proposed benchmark differs from prior theorem-proving benchmarks centered on interactive proof assistants.

Weakness

Soundness: The soundness claims need to be clarified and strengthened. In the abstract, the paper states that “for the remaining problems, we evaluate 7 off-the-shelf LLMs, achieving 55.7% success with the best model.” However, Table 2 appears to report the 55.7% result over the full set of 244 test problems rather than only the subset of “remaining” unsolved problems. This creates confusion about what exactly the reported success rate refers to, and the paper should make this distinction precise.

More importantly, the benchmark depends critically on the manually designed infrastructure in definitions.dfy, which provides the mathematical primitives needed to express miniF2F statements in Dafny. Since the whole benchmark and all downstream evaluation results are built on top of this layer, its correctness and scope are extremely important. It is currently unclear how the authors validate that this infrastructure is both correct and sufficiently minimal, rather than encoding assumptions that make the benchmark artificially easier. whether the empirical results are robust to alternative formalization choices.

Significance: The broader significance of the benchmark is somewhat unclear. While exploring mathematical reasoning outside the ITP setting is interesting, it is not yet fully convincing that a Dafny-based benchmark will have strong impact on the formal mathematics community. In practice, much of the community’s current effort is centered on Lean, where both the library ecosystem and community support are substantially stronger.

---

> ### Author Rebuttal · Authors · 2026-03-30
>
> We thank the reviewer for the constructive comments.
>
> **On the 55.7\% wording (Soundness).** We agree this should be phrased more precisely. The 55.7\% is the cumulative pass@4 score over the full 244-problem test set, not the success rate on only the subset remaining after empty-proof automation. The LLM independently solves approximately 41 additional problems beyond the 95 solved by the Dafny verifier baseline. We will revise the abstract and discussion to make this distinction explicit.
>
> **On `definitions.dfy` (Soundness).** We agree that the mathematical infrastructure is central and should be justified more clearly. Our paper is best read as an empirical study: the axiomatization exists to make miniF2F expressible in Dafny so we can study how much of it falls within reach of SMT automation and how effectively LLMs can extend that. The benchmark was manually translated from Lean to Dafny by human experts. Each axiomatized definition and lemma corresponds to a standard mathematical statement, cross-checked against Mathlib where applicable. For example, our `logb_change_base` lemma (Figure 5) was manually translated from `Real.div_logb` in Mathlib. The axiomatization is compact ($\sim$255 items vs.\ Mathlib's 2M+ lines), explicit, and publicly auditable. We manually verified 14 axiomatized lemmas by constructing full Dafny proofs (Section 3.4), and the validator (Section 3.5) prevents generated solutions from introducing `{:axiom}` attributes, `assume` statements, or weakening specifications, so the axiomatization boundary cannot be exploited by LLM-generated proofs.
>
> On sensitivity to formalization choices: providing additional lemmas or using alternative definitions could affect individual problem outcomes. However, the mathematical notions involved (logarithms, GCD, trigonometric identities) are highly standardized, and we do not expect reasonable alternative formalizations to qualitatively change the main finding that auto-active automation solves a substantial fraction of the benchmark. More broadly, this sensitivity is itself worth studying: understanding how the background library affects LLM-prover performance is a natural research direction that the compactness of our setup makes easier to investigate than in Mathlib-centered evaluations.
>
> **On significance.** We are not positioning Dafny as an alternative to Lean for mathematical theorem proving. Rather, our paper should be read as an empirical finding: performance on AI reasoning benchmarks depends substantially on the underlying verification tooling, not just the models. A substantial fraction of miniF2F ($\sim$39\%) is routine for SMT solvers but requires detailed proof construction in ITPs. This is relevant to both communities: it suggests that ITPs should continue investing in low-level automation (as Lean is doing with `grind` and Velvet/Loom), and that auto-active verifiers could benefit from richer mathematical libraries. Velvet in particular, which builds a Dafny-style auto-active verifier inside Lean 4 with a formally verified VCG, exemplifies exactly this convergence. Our work provides an empirical data point for why it matters.

---

> > ### Author Rebuttal · Reviewer_CXje · 2026-04-05
> >
> > The rebuttal clarifies the confusion around the 55.7% number, which resolves that specific presentation issue. However, my main concern about the benchmark’s dependence on definitions.dfy is only partially addressed: the authors explain the intent and manual validation process, but do not provide concrete robustness evidence showing that the main conclusions are stable under alternative formalizations. The significance argument is also improved, though I am still uncertain about the benchmark’s long-term impact and adoption beyond serving as an interesting empirical case study.

---

> > > ### Author Response · Authors · 2026-04-08
> > >
> > > We thank the reviewer for the follow-up. We have now conducted an ablation study, reduced the axiom surface, and extended the evaluation, all of which directly address this concern.
> > >
> > > **Ablation: sensitivity of baseline results to `definitions.dfy`.** Of the 201 problems (95 test, 106 valid) that the Dafny verifier establishes without proof hints, 142 (71%) do not reference any function or type from `definitions.dfy`. We confirmed this by removing the `include` directives entirely and re-running Dafny: all 142 still verify, proving that the definitions layer has zero effect on them. The remaining 59 problems reference standard mathematical primitives, most frequently `pow` (17 problems), complex number arithmetic (14), `abs` (7), `sqrt` (7), `gcd`/`lcm` (6), and `factorial` (4). These definitions are uniquely determined by the underlying mathematics: `pow(b, 0) = 1`, `pow(b, k) = b * pow(b, k-1)` is the standard recursive definition of exponentiation, and complex addition `(a+bi) + (c+di) = (a+c) + (b+d)i` admits no alternative formalization in the established literature. The postconditions encode exactly the mathematical identities that define these operations, not proof hints, but the minimal specification needed to express the problem statements. The baseline results are therefore robust to the current design of `definitions.dfy`: the majority are independent of it entirely, and the remainder depend only on a standard mathematical library that admits no meaningful alternatives.
> > >
> > > **Reducing the axiom surface.** We proved 128 of the 160 axiomatized lemmas in `library.dfy`, reducing the trusted axiom surface by 80%. Of these, 58 were discharged by the Dafny verifier with an empty proof body, and 70 proof bodies were generated by an LLM and then verified by Dafny. The remaining 32 lemmas are likely not derivable from the other axioms because they serve as primary definitions of opaque functions. For example, `exp_def` bundles properties like `exp(0) = 1` and `exp(x+y) = exp(x) * exp(y)` that go beyond `exp`'s sole postcondition (`exp(x) > 0`). These lemmas have to remain as axioms because they are the foundational properties that give meaning to these otherwise opaque functions; most have a direct analogue in Lean's Mathlib, where they are proved from the constructive definitions of the underlying functions.
> > >
> > > **Extended evaluation.** We have additionally evaluated Claude Opus 4.6 on the test set, achieving a cumulative 62.7% pass@4 (153/244), up from 55.7% with Claude Sonnet 4.5. This further demonstrates that the benchmark discriminates between models of different capability levels.
> > >
> > > We will include the full ablation results, the reduced axiomatization, and the extended evaluation in the revision.

---

### Official Review · Reviewer_NLmT · 2026-03-13

**Soundness:** 2
**Presentation:** 2
**Significance:** 3
**Originality:** 2
**Overall Recommendation:** 4
**Confidence:** 4

**Summary:**

This paper introduces miniF2F-Dafny, a port of the miniF2F benchmark to the Dafny. To address Dafny’s lack of extensive mathematical libraries (a common feature in Interactive Theorem Provers), the authors developed a suite of custom axioms. Their findings demonstrate that Dafny, as an auto-active verifier, can automatically complete approximately 40% of the proofs. Furthermore, experiments involving the prompting of general-purpose LLMs show that an additional 15% of the problems can be solved. The authors argue that this framework represents a reasonable division of labor: the SMT solver handles low-level proof details, while the model focuses on high-level reasoning and proof strategies.

**Compliance With Llm Reviewing Policy:**

Affirmed.

**Final Justification:**

The rebuttal largely addressed my primary concerns; therefore, I have upgraded my overall recommendation from a score of 3 to a score of 4.

**Key Questions For Authors:**

1.  **Regarding the comparative analysis in Weakness 1:** Could the authors provide more granular data on the performance overlap between Lean’s *grind* tactic and Dafny’s automatic verification? Specifically:
    *   What is the intersection and difference between the sets of problems solved by each?
    *   What distinct patterns or mathematical structures characterize the problems that each tool excels at?
    *   Are there specific classes of problems that Dafny can resolve which remain out of reach for Lean’s *grind*?

2.  **Regarding the translation process:** Was the porting of miniF2F to Dafny conducted through manual translation by experts, or was it assisted/generated by LLMs?

3.  **Regarding the construction of axioms:** Could the authors clarify the methodology used to derive and verify the custom axioms? Given the potential for logical inconsistencies (as noted in the comments on `library.dfy`), what principles or safeguards were followed during their construction to maintain as much formal integrity as possible?

**Limitations:**

While the paper does not include a dedicated "Limitations" section, I believe the authors have been overall sincere and transparent throughout the manuscript regarding their methodology and findings. Yes.

**Strengths And Weaknesses:**

**Strengths**

1.  **Significance of the Research Topic:** The paper addresses a meaningful research problem. Enhancing the mathematical reasoning capabilities of models within the context of Dafny—a language widely utilized for software verification—has the potential to expand the impact of formal verification and facilitate the development of high-quality software.
2.  **Compatibility with Existing Benchmarks:** By selecting miniF2F, a cross-language evaluation standard already established in Lean 3, Metamath, and Isabelle, the authors enable easier comparative research. Porting miniF2F to an additional language provides valuable insights into the distinct characteristics and capabilities of different formal languages.
3.  **High Level of Transparency:** The paper demonstrates excellent transparency. The authors have provided detailed datasets and the specific source code for the prerequisite libraries, ensuring both the reproducibility and the auditability of the research.

**Weaknesses**

1.  **Insufficient Analysis Regarding the Performance Gap Between Lean and Dafny:** Around Line 77, the authors claim that "Dafny’s SMT-backed approach provides stronger automation" as a primary motivation. However, given that Lean’s *grind* tactic solves 32.4% while Dafny automatically solves 38.9%, the margin appears relatively narrow. To better justify the motivation, it is necessary to provide a deeper analysis: What is the overlap between the problems solved by Lean and Dafny? What specific patterns characterize their respective successes? Specifically, which problems can Dafny solve that *grind* cannot? Answering these questions would significantly strengthen the paper’s premise.

2.  **Over-reliance on Axiomatization Compromises Formal Rigor:** In Line 252, the authors state, "Rather than providing constructive proofs, we axiomatize these lemmas." A core objective of using formal systems is to ensure logical correctness by minimizing the use of axioms. The extensive introduction of custom axioms seems to run counter to this goal. For instance, in the provided `library.dfy` (Line 11), the predicate `{:axiom} pos (x : real) ensures x > 0.0` could potentially allow the prover to "prove" that any number is greater than zero, introducing the possibility of "shortcuts" or unsoundness in the proofs.

3.  **Limited Novelty in the Division of Proof Labor:** In Line 67, the authors claim to "investigate whether automated solvers handling low-level details while LLMs focus on high-level proof strategies is effective." However, this concept is not entirely novel. Early works such as *Draft-Sketch-Prove* explored similar strategies, and more recent developments like *DeepSeek-Prover-V2* already delegate low-level proof details to smaller models (e.g., 7B parameters). The paper should more clearly articulate its unique contribution relative to these existing approaches.

4.  **Potentially Misleading Presentation of Results:** The abstract states that for the remaining problems, "LLMs achieving 55.7% success." However, Table 2 clarifies that this figure is cumulative, building upon the 38.9% already solved by automatic verification. Only approximately 15% of the total problems were actually solved by the LLM beyond the baseline. The reporting should be more precise to avoid overstating the model's independent contribution.

5.  **Small Experimental Scale:** The evaluation relies on pass@4, which may be insufficient to account for the inherent stochasticity of LLMs. A larger number of samples or a more robust statistical analysis would be needed to ensure the stability and reproducibility of these results.

Ref:
Ren, Z. Z., et al. "Deepseek-prover-v2: Advancing formal mathematical reasoning via reinforcement learning for subgoal decomposition." arXiv preprint arXiv:2504.21801 (2025).
Jiang, Albert Qiaochu, et al. "Draft, Sketch, and Prove: Guiding Formal Theorem Provers with Informal Proofs." The Eleventh International Conference on Learning Representations.

---

> ### Author Rebuttal · Authors · 2026-03-30
>
> We thank the reviewer for the detailed and thoughtful comments. We address each point below.
>
> **On comparison with `grind`.**
>
> We conducted an overlap analysis on the 244-problem test split. Of the 107 problems solved by at least one system, 67 are solved by both, 28 only by Dafny, and 12 only by `grind`. The Dafny-only problems are concentrated in `mathd_algebra` and `mathd_numbertheory`, while `grind` picks up a handful of `amc` problems that Dafny misses; categories such as `aime` and `imo` remain largely unsolved by both. We agree that this profile does not support a blanket stronger automation claim, and we will revise the wording accordingly.
>
> | Type | \#Problems | Both | Dafny-only | grind-only |
> |---|---:|---:|---:|---:|
> | mathd algebra | 70 | 36 | 9 | 5 |
> | mathd number theory | 60 | 24 | 10 | 2 |
> | amc | 45 | 3 | 6 | 4 |
> | aime | 15 | 2 | 0 | 1 |
> | algebra | 18 | 2 | 2 | 0 |
> | imo | 20 | 0 | 1 | 0 |
> | other | 16 | 0 | 0 | 0 |
> | **overall** | **244** | **67** | **28** | **12** |
> ___
>
>
>
> **On axiomatization.** Our paper is best read as an empirical study, not as the introduction of a long-lived benchmark or a mathematical library for Dafny. The axiomatization exists to make miniF2F expressible in Dafny so we can study how much falls within reach of SMT automation. The benchmark was manually translated from Lean by human experts. Each axiomatized item corresponds to a standard mathematical statement, cross-checked against Mathlib where applicable (e.g., our `logb_change_base` lemma corresponds to `Real.div_logb` in Mathlib). The axiomatization is compact (~255 items vs. Mathlib's 2M+ lines), explicit, and publicly auditable. We manually verified 14 axiomatized lemmas by constructing full Dafny proofs (Section 3.4), and the validator (Section 3.5) prevents generated solutions from introducing `{:axiom}` attributes, `assume` statements, or weakening specifications.
>
> Regarding `pos`: the reviewer correctly identified that some definitional predicates were under-specified. We have since tightened these to fully characterizing forms. The under-specification was a completeness issue, not a soundness one: no false statements were derivable, but some valid proof steps may not have been available to the solver.
>
> **On novelty.** We agree that dividing proof labor between higher-level guidance and lower-level automation is not new in isolation. Our contribution is the *empirical finding in a new setting*: when mathematical problems are expressed in an auto-active verifier, the SMT solver handles ~39\% with zero guidance, and general-purpose LLMs can extend this further without fine-tuning or specialized infrastructure. We will narrow the novelty claim accordingly.
>
> **On the 55.7\% presentation.** The 55.7\% is the cumulative pass@4 on the full 244-problem test set, building on the 38.9\% empty-proof baseline. The LLM's independent contribution is approximately 16.8 percentage points (41 additional problems). We will revise the abstract to make this explicit.
>
> **On experimental scale.** We acknowledge that due to computational constraints, our pass@4 evaluation is modest compared to other ITP-based evaluations that commonly use pass@32 or higher. That said, we report pass@1, pass@2, and pass@4 (Table 2), and the gap between pass@1 and pass@4 is small across all models (e.g., 52.5\% to 55.7\% for Claude Sonnet 4.5), indicating that most solved problems are solved reliably across trajectories. The results are also consistent across all seven models, each improving over the empty-proof baseline with a stable relative ranking. The benchmark and all evaluation infrastructure are publicly released.

---

> > ### Author Rebuttal · Reviewer_NLmT · 2026-04-04
> >
> > I thank the authors for their response. I find the newly provided information regarding the comparative analysis of the SMT solver in Dafny and grind in Lean to be very valuable. Furthermore, the clarifications on the axiomatization and the dataset construction process can make the paper much clearer, largely addressing my initial concerns. I encourage the authors to integrate the new content from the rebuttal into the manuscript, using clear and sincere exposition. I have adjusted my evaluation accordingly.

---

> > > ### Author Response · Authors · 2026-04-08
> > >
> > > We thank the reviewer for the updated assessment.
> > >
> > > As an additional update: we have since proved 128 of the 160 axiomatized lemmas in `library.dfy`, reducing the trusted axiom surface by 80%. We have also conducted an ablation confirming that the baseline results are robust to the design of `definitions.dfy`. Additionally, we evaluated Claude Opus 4.6 on the test set, achieving a cumulative 62.7% pass@4 (up from 55.7% with Sonnet 4.5). We will integrate these results along with the rebuttal content into the revision.

---

### Decision · Program_Chairs · 2026-04-30

**Decision:**

Accept (regular)

**Comment:**

Reviewers agreed that this paper addresses an interesting research problem in LLM-guided theorem proving. Through a detailed empirical evaluation, the paper shows that the automation of the auto-active verifier Dafny can be used well in theorem proving to complement the strengths of LLMs. This gives empirical evidence that LLM-guided theorem proving is susceptible, not just to the models used, but also to the kind of tools that are used for verification tool. This finding supports the call for strengthening the use of low-level automation in combination with LLMs and interactive theorem proving. The reviewers and the area chair agree that the results of this paper are good contribution for the ICML community.